# Measuring Pointwise $\mathcal{V}$-Usable Information In-Context-ly

**Sheng Lu[1], Shan Chen[2,3], Yingya Li[4],**
**Danielle Bitterman[2,3], Guergana Savova[4],** and **Iryna Gurevych[1]**

[1] Ubiquitous Knowledge Processing Lab (UKP Lab), Technical University of Darmstadt
[2] Artificial Intelligence in Medicine (AIM) Program, Mass General Brigham, Harvard Medical School
[3] Department of Radiation Oncology, Brigham and Women's Hospital/Dana-Farber Cancer Institute
[4] Computational Health Informatics Program, Boston Children's Hospital, Harvard Medical School

www.ukp.tu-darmstadt.de

## Abstract

In-context learning (ICL) is a new learning paradigm that has gained popularity along with the development of large language models. In this work, we adapt a recently proposed hardness metric, pointwise $\mathcal{V}$-usable information (PVI), to an in-context version (in-context PVI). Compared to the original PVI, in-context PVI is more efficient in that it requires only a few exemplars and does not require fine-tuning. We conducted a comprehensive empirical analysis to evaluate the reliability of in-context PVI. Our findings indicate that in-context PVI estimates exhibit similar characteristics to the original PVI. Specific to the in-context setting, we show that in-context PVI estimates remain consistent across different exemplar selections and numbers of shots. The variance of in-context PVI estimates across different exemplar selections is insignificant, which suggests that in-context estimates PVI are stable. Furthermore, we demonstrate how in-context PVI can be employed to identify challenging instances. Our work highlights the potential of in-context PVI and provides new insights into the capabilities of ICL.[1]

## 1 Introduction

Understanding the hardness of a dataset or an instance is crucial for understanding the progress in machine learning, since a dataset is designed as a proxy for real-world tasks (Torralba and Efros, 2011). The significance of hardness has been acknowledged in the field of natural language processing (NLP) (Hahn et al., 2021; Perez et al., 2021; Zhao et al., 2022; Gadre et al., 2023). Extended from the *predictive $\mathcal{V}$-information* framework (Xu et al., 2020), pointwise $\mathcal{V}$-usable information (PVI) is a recently proposed metric for measuring the hardness of individual instances (Ethayarajh et al., 2022). PVI is estimated through supervised learning, which involves fine-tuning two models: one

model that is fine-tuned on *input-target* pairs, and another model fine-tuned on only the target labels. PVI measures the amount of *usable* information in an input for a given model, which reflects the ease with which a model can predict a certain label given an input. Though it is a recently proposed method, the effectiveness of PVI has been demonstrated in various NLP tasks (Chen et al., 2022a; Kulmizev and Nivre, 2023; Lin et al., 2023; Prasad et al., 2023).

Recent years have seen remarkable progress in the development of large language models (LLMs), and it is expected that they will continue to be an essential topic in NLP (Brown et al., 2020; Chung et al., 2022; Chowdhery et al., 2022; Touvron et al., 2023). In the era of LLMs, PVI is useful in many aspects as a measure of hardness, such as the development of high-quality new benchmarks that can better evaluate the capabilities of LLMs and the selection of in-context exemplars that enhance model performance. However, given the scales of the state-of-the-art LLMs, the calculation of PVI can be challenging due to the need for fine-tuning. Motivated by the need to leverage PVI for LLMs and the recent discoveries that in-context learning (ICL) is similar to fine-tuning (Akyürek et al., 2022; von Oswald et al., 2022; Dai et al., 2022), we implement PVI in an in-context manner (i.e., in-context PVI). Rather than fine-tuning, we prompt a model using two few-shot prompts, an *input-target* prompt and a *target-only* prompt (see A.2 for an example). Figure 1 shows the difference between the calculation of PVI and in-context PVI.

This work aims to investigate the feasibility of obtaining reliable PVI estimates in an in-context setting. We conducted experiments with various datasets and a range of LLMs. We showed that in-context PVI behaves similarly to the original PVI: in-context PVI estimates are consistent across different models, and the threshold at which predictions become incorrect is similar across datasets.

---

[1] Our code is available at https://github.com/UKPLab/in-context-pvi.

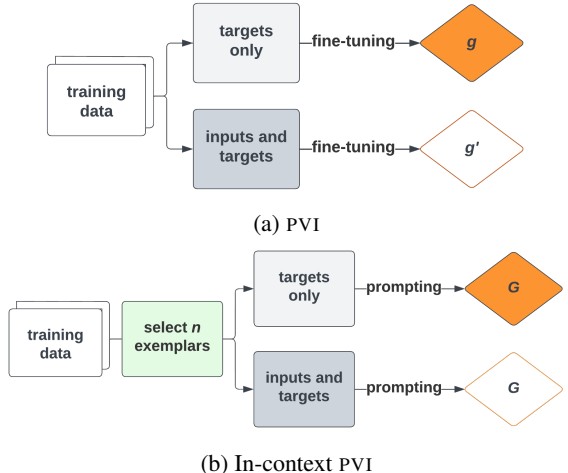

(a) PVI

(b) In-context PVI

Figure 1: The main difference between the calculation of PVI and in-context PVI. *g* and *g'* are two fine-tuned models and *G* is an LLM.

Specific to the in-context setting, we found that in-context PVI estimates are consistent across different selections of exemplars and numbers of exemplars in the prompt. We also found that the variance of in-context PVI estimates across different exemplar selections is insignificant, which suggests that in-context PVI estimates are quite stable. We assessed the correlation between in-context PVI estimates and inter-annotator agreement, and our findings indicate that in-context PVI estimates made by larger models are more reliable. We conducted a qualitative analysis to show how in-context PVI assists in identifying challenging instances. Additionally, we presented a preliminary analysis on the potential of in-context PVI to improve model performance through exemplar selection.

Our contributions are summarized as follows:

- We propose in-context PVI, a new approach to calculating PVI which is more efficient than the original method.

- We present an extensive empirical analysis to demonstrate the reliability of in-context PVI.

- Our work provides new insights into the capabilities of ICL.

## 2 Related work

### 2.1 In-context learning

In-context learning (ICL) is a popular learning approach that emerges alongside the advent of LLMs (Brown et al., 2020; Liu et al., 2023). It typically involves using several demonstrations in natural language, which provides a more interpretable interface and greatly reduces the computation costs compared to supervised training (Dong et al., 2022). ICL has shown strong performance on a series of natural language tasks (Kojima et al., 2022; Saparov and He, 2022; Srivastava et al., 2022; Wei et al., 2022a,b).

Recent studies have enhanced our understanding of ICL and its mechanism. Akyürek et al. (2022) and von Oswald et al. (2022) investigate ICL in regression problems, and show that transformer-based in-context learners implement gradient descent in an implicit way, which can be related to gradient-based meta-learning formulations. Similarly, Dai et al. (2022) argue that ICL performs implicit fine-tuning, and understand ICL as a process of meta-optimization. A transformer-based model is a meta-optimizer which produces meta-gradients according to the exemplars through forward pass. These meta-gradients are applied to the original model through attention. Dai et al. (2022) provide empirical evidence that ICL behaves similarly to fine-tuning at the prediction, representation, and attention levels.

One of the most serious concerns in ICL is that in-context learners are reported to be very sensitive to changes in the input (Liu et al., 2022; Lu et al., 2022; Zhao et al., 2021; Chang and Jia, 2023; Chen et al., 2022b; Wang et al., 2023). The reliability of an ICL related method is significantly reduced if the output changes drastically with the use of another set of exemplars. Therefore, it is critical to ensure that an ICL related method maintains consistency across varying exemplar selections.

### 2.2 Hardness

Hardness refers to the difficulty of an instance in a given distribution or the difficulty of a dataset for a given model (Ethayarajh et al., 2022). It is an important concept to understand the quality of a dataset (Torralba and Efros, 2011; Zhao et al., 2022; Cui et al., 2023). In addition, the concept of hardness is crucial to the study of human-AI interaction, where hardness estimates are essential to evaluate each AI agent's capabilities and facilitate more effective collaboration (Spitzer et al., 2023).

Pointwise $\mathcal{V}$-usable information (PVI) is a recently proposed metric that measures the hardness of an instance (Ethayarajh et al., 2022). PVI is based on the *predictive $\mathcal{V}$-information* framework, which incorporates mutual information and other

types of informativeness such as the coefficient of determination (Xu et al., 2020). Ethayarajh et al. (2022) extended this framework by framing dataset or instance difficulty as the lack of $\mathcal{V}$-usable information. A high PVI estimate indicates that the input is well represented in the model, and thus the instance is considered to be easier for the model. A low PVI estimate indicates that the input contains little information, so the instance is considered to be harder. PVI allows us to compare the hardness of different instances, or the hardness of subsets by computing the average PVI over the instances.

Although it is a recently proposed metric, PVI has received considerable attention and has proven to be effective in various tasks. It is used as a quality estimate of Universal Dependencies treebanks in Kulmizev and Nivre (2023). PVI is used to select synthetic data as an augmentation to an intent detection classifier, which achieves state-of-the-art performance (Lin et al., 2023). Chen et al. (2022a) and Prasad et al. (2023) incorporate PVI into an informativeness metric to evaluate rationales, and find that it captures the expected flow of information in high-quality reasoning chains.

## 3 Method

Algorithm 1 shows the calculation of PVI, which involves fine-tuning a model $\mathcal{G}$ on two different datasets.

---

**Algorithm 1** The calculation of PVI

---

1: **Input:** a dataset $\mathcal{D}$, a model $\mathcal{G}$, a test instance $(x, y)$
2: $g' \leftarrow$ fine-tune $\mathcal{G}$ on $\mathcal{D}$
3: $g \leftarrow$ fine-tune $\mathcal{G}$ on $\{(\varnothing, y_i) | (x_i, y_i) \in \mathcal{D}\}$
4: $\text{PVI}(x, y) \leftarrow -\log_2 g[\varnothing](y) + \log_2 g'[x](y)$

---

In Algorithm 1, $g'$ is fine-tuned on $\mathcal{D}$, i.e., the *input-target* pairs $\{(x_i, y_i) | (x_i, y_i) \in \mathcal{D}\}$, and $g$ is fine-tuned on *null-target* pairs $\{(\varnothing, y_i) | (x_i, y_i) \in \mathcal{D}\}$ ($\varnothing$ is an empty string). PVI is interpreted as the information gain when an input is provided to fine-tune $\mathcal{G}$.

In-context pointwise $\mathcal{V}$-usable information (in-context PVI) is adapted from the original PVI. Instead of fine-tuning a model $\mathcal{G}$, two few-shot prompts, i.e., an *input-target* prompt $p' = (x_1, y_1, x_2, y_2, ..., x_n, y_n, x)$ and a *null-target* prompt $p = (\varnothing, y_1, \varnothing, y_2, ..., \varnothing, y_n, \varnothing)$, are used to prompt $\mathcal{G}$. In-context PVI, denoted as $C(x, y)$,

is calculated as Equation (1):

$$C(x, y) = -\log_2 \mathcal{G}[p](y) + \log_2 \mathcal{G}[p'](y). \quad (1)$$

Given the observation that ICL resembles fine-tuning (Dai et al., 2022), $\log_2 \mathcal{G}[p](y)$ and $\log_2 \mathcal{G}[p'](y)$ are the ICL approximations of $\log_2 g[\varnothing](y)$ and $\log_2 g'[x](y)$ in Algorithm 1.

The calculation of the original PVI is based on sequence classification, i.e., the dimension of the output space of $g'$ and $g$ is the number of unique labels in a task. Different to the original PVI, the calculation of in-context PVI is based on generation, where the output is a sequence of tokens such as ["un", "acceptable"]. Instead of asking the model to produce a label prediction such as "unacceptable", we matched a numerical index to a label as the target in $p'$ and $p$ (see A.2 for an example), so that the expected output is a single token, which makes the calculation of $\log_2 \mathcal{G}[p](y)$ and $\log_2 \mathcal{G}[p'](y)$ easier.

## 4 Experiment settings

**Dataset** We conducted experiments on 7 datasets, including BoolQ (Clark et al., 2019), CoLA (Warstadt et al., 2019), MultiNLI (Williams et al., 2018), SNLI (Bowman et al., 2015), RTE (Wang et al., 2019), and two domain specific datasets–Health Advice (Li et al., 2021), and Causal Language (Yu et al., 2019). The selected datasets cover question answering, natural language inference and reasoning in both general and domain specific settings. We included the three tasks that are used in Ethayarajh et al. (2022), i.e., CoLA, MultiNLI, and SNLI. BoolQ and RTE are two commonly used datasets from SuperGlue (Wang et al., 2019). Health Advice and Causal Language involve fundamental tasks in the biomedical domains, which require domain specific knowledge. The details of the datasets are shown in Tables 9, 10, and 11 in A.1.

**Model** We tested models with sizes varying from 125M to 175B: GPT2-125M (Radford et al., 2019), GPT-Neo-1.3B, GPT-Neo-2.7B (Gao et al., 2020), GPT-J-6B, GPT-JT-6B (Wang and Komatsuzaki, 2021), Alpaca-7B, Alpaca-13B (Taori et al., 2023), and OpenAI text-davinci-003 (GPT3-175B). We limited the use of GPT3-175B to experiments on CoLA, RTE, and the first 500 test instances in MultiNLI and SNLI due to cost considerations.

**Exemplar selection** We used 3 sets of exemplars which were randomly selected from the training

| dataset | accuracy | acc. low PVI | acc. high PVI | PVI for TRUE | PVI for FALSE |
|---------|----------|--------------|---------------|--------------|---------------|
| CoLA | 0.7242 | 0.0000 | 1.0000 | 1.1640 | -7.6833 |
| MultiNLI | 0.7322 | 0.0000 | 0.9738 | 2.5702 | -3.5963 |
| RTE | 0.7401 | 0.0417 | 0.9345 | 1.7554 | -3.9994 |
| SNLI | 0.6704 | 0.0000 | 0.9967 | 2.5118 | -4.7711 |

Table 1: The prediction accuracy, the average prediction accuracy for instances with the lowest 20% in-context PVI estimates (acc. low PVI), the average prediction accuracy for instances with the highest 20% in-context PVI estimates (acc. high PVI), average in-context PVI estimates for correct predictions (PVI for TRUE), and average in-context PVI estimates for incorrect predictions (PVI for FALSE) for runs using GPT3-175B. Statistics are averaged over the results obtained using 3 sets of exemplars. Please refer to Table 14 in A.3 for more results.

set. See A.2 for examples of the prompts we used.
**Number of shots** We tried different numbers of shots. The minimum number of shots is equal to the number of unique labels in a dataset. Apart from that, we also tried a number of shots that is twice the minimum number of shots.[2] For instance, there are three unique labels in MultiNLI, so we tried 3- and 6-shot on it. We did not consider methods that scale up in-context exemplars such as structured prompting (Hao et al., 2022), since we want to test in-context PVI in a typical in-context setting, i.e., a few-shot setting (Dong et al., 2022).

## 5 Results

Table 1 provides an overview of in-context PVI estimates across datasets, along with their relationship with prediction accuracy. The prediction accuracy for instances with low in-context PVI estimates is lower than that for instances with high in-context PVI estimates, and the average in-context PVI estimate for correct predictions is higher than that for incorrect predictions. Note that the interpretation of in-context PVI is relative, which means that a single value on its own is not as important as comparing a range of in-context PVI estimates. Please refer to Table 14 in A.3 for more results.

Figure 2 shows the difference in prediction accuracy between instances with the top 20th/50th percentile and bottom 20th/50th percentile of in-context PVI estimates. We observed an increase in accuracy gain when model scales with one exception (GPT2-125M).

### 5.1 The consistency of in-context PVI

Table 2 shows the consistency of in-context PVI estimates across different selections of exemplars and numbers of shots, which are two essential features for in-context learning.

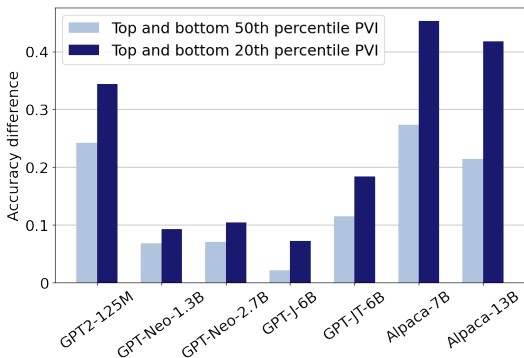

Figure 2: The difference in prediction accuracy between instances with the top 20th/50th percentile and bottom 20th/50th percentile in-context PVI estimates. Statistics are averaged over the results obtained using 7 datasets, 3 sets of exemplars, and 2 numbers of shots.

| variable | avg. | med. | $r > 0.6$ | $r < 0.3$ |
|----------|------|------|-----------|-----------|
| Exemplars | 0.55 | 0.71 | 59.4% | 22.9% |
| Shots | 0.44 | 0.74 | 57.6% | 29.9% |

Table 2: The average (avg.) and median (med.) *Pearson* correlation coefficients of in-context PVI estimates obtained using different sets of exemplars (Exemplars) and numbers of shots (Shots). We also report the percentages of cases where there is a strong correlation ($r > 0.6$) and weak correlation ($r < 0.3$).

The results show that there is a moderate to strong correlation between in-context PVI estimates obtained using different sets of exemplars, with an average *Pearson* correlation of 0.55 and a median of 0.71. In fact, 59.4% of the cases showed a strong correlation with a *Pearson* coefficient above 0.6. We observed a stronger correlation between in-context PVI estimates across different numbers of shots, with an average *Pearson* coefficient of 0.44 and a median of 0.74. Nearly 60% of the cases showed a *Pearson* coefficient over 0.6, which suggests a substantial level of consistency.[3]

---

[2]Since CoLA contains only 2 labels and the input sentences are short, we tried 4- and 8-shot for it.

[3]Please see full results in https://github.com/UKPLab/in-context-pvi for the corresponding *p*-values of the statistics in Table 2.

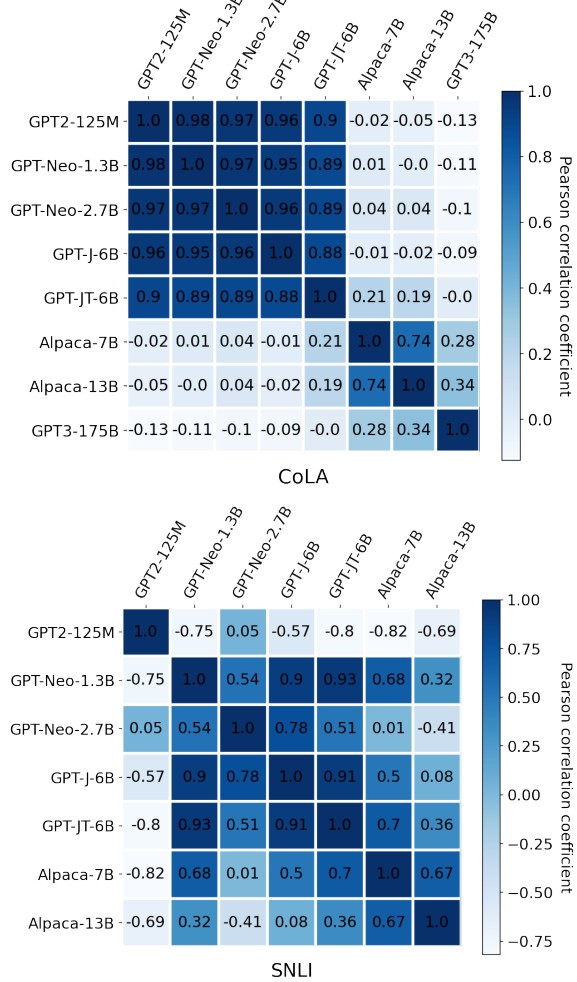

Figure 3: The *Pearson* correlation coefficients among in-context PVI estimates made by different models for SNLI and CoLA instances. In-context PVI estimates are obtained using a 4-shot prompt for CoLA and a 6-shot prompt for SNLI.

Figure 3 shows the *Pearson* correlation coefficients between in-context PVI estimates for CoLA and SNLI instances across different models. We noted that in-context PVI estimates made by GPT-Neo-1.3B, GPT-Neo-2.7B, GPT-J-6B, and GPT-JT-6B, and those made by and Alpaca-7B and Alpaca-13B are generally much more consistent than, for example, those made by GPT2-125M and Alpaca-13B (see A.4 for more examples). This is probably due to certain models being *closer* to one another in terms of their architecture, training data, etc. Also, instances that are challenging for smaller models may not pose similar challenges to larger models (Srivastava et al., 2022; Wei et al., 2022a), which explains the inconsistency between in-context PVI estimates made by larger models, such as GPT3-175B, and those made by smaller models.

Overall our results point towards a considerable degree of consistency in in-context PVI estimates across different exemplar selections and numbers of shots. There are no significant changes in in-context PVI estimates given different sets of exemplars, which is critical for the practical application of in-context PVI. We also observed that in-context PVI estimates are much more consistent among *closer* models, i.e., those having similar architectures, training data, etc.

## 5.2 The variance of in-context PVI estimates across different sets of exemplars

We used one-way analysis of variance (ANOVA) to examine the variance of in-context PVI estimates across different exemplar selections. Our hypothesis is that in-context PVI estimates obtained using different sets of exemplars have similar means. Table 3 shows a part of the results (please refer to A.5 for the rest of the results):

| dataset | model | *F*-statistic | *p*-value |
|---------|-------|---------------|-----------|
|         | GPT2-125M | 0.2958 | 0.7634 |
|         | GPT-neo-1.3B | 0.5543 | 0.6239 |
|         | GPT-neo-2.7B | 0.0383 | 0.9629 |
| MultiNLI | GPT-J-6B | 0.2604 | 0.7866 |
|         | GPT-JT-6B | 0.3099 | 0.7545 |
|         | Alpaca-7B | 0.0363 | 0.9647 |
|         | Alpaca-13B | 0.4304 | 0.6850 |

Table 3: The results of one-way ANOVA for the variance of in-context PVI estimates of MultiNLI instances across different sets of exemplars.

The results reveal that there are no statistically significant differences ($p$-value $> 0.05$) in in-context PVI estimates obtained using different sets of exemplars. *F*-statistics also show that the between-group difference of in-context PVI estimates among the three sets of exemplars is smaller than the difference within each set. In other words, in-context PVI estimates are quite stable.

## 5.3 In-context PVI threshold for incorrect predictions

Figure 4 displays the density distribution of in-context PVI estimates made by the smallest and largest model we tested, i.e., GPT2-125M and GPT3-175B, for correctly and incorrectly predicted instances. Statistics in Figure 4 are based on in-context PVI estimates obtained using one randomly selected set of exemplars, and in-context PVI estimates made by GPT2-125M were obtained using more exemplars than those made by GPT3-175B. Similar to what Ethayarajh et al. (2022) report,

high in-context PVI instances are more likely to be predicted correctly while low in-context PVI instances are not, and the threshold at which instances start being predicted incorrectly is similar across datasets. However, in-context PVI estimates made by GPT2-125M (see Figure 4a) are much noisier than those made by GPT3-175B (see Figure 4b). This suggests that in-context PVI estimates made by larger models better capture information that is useful to produce the correct predictions.

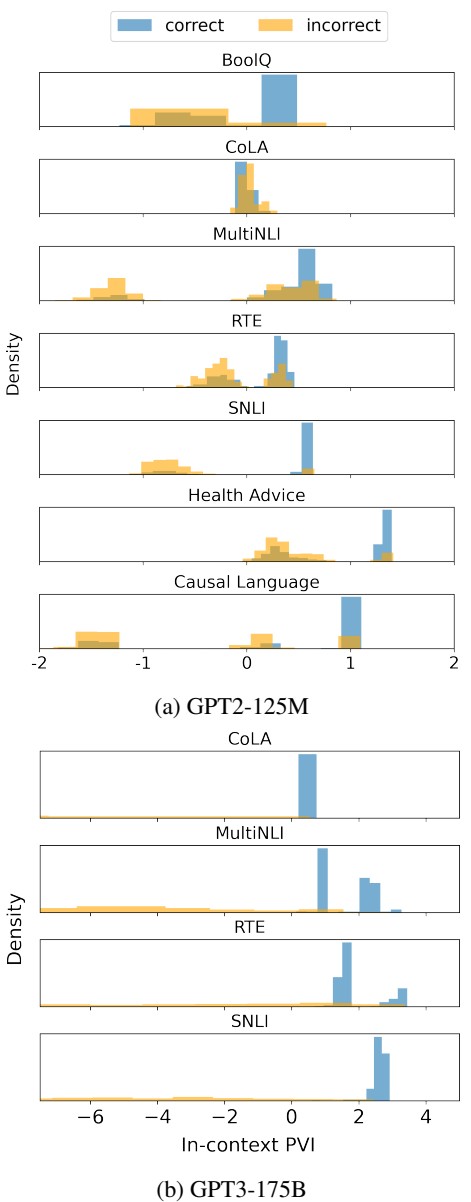

(a) GPT2-125M

(b) GPT3-175B

Figure 4: The density distribution of in-context PVI estimates made by GPT2-125M and GPT3-175B for correctly and incorrectly predicted instances in terms of datasets. GPT3-175B results for MultiNLI and SNLI are based on the first 500 test instances.

## 5.4 Correlation with inter-annotator agreement

MultiNLI (Williams et al., 2018) and SNLI (Bowman et al., 2015) both contain annotations made by five annotators. The difference in human annotations can be viewed as an indicator of instance hardness: the less the annotators agree, the harder an instance is. We adopted a straightforward measure of agreement, variation-ratio (Freeman, 1965), and measured inter-annotator agreement as the frequency of the most frequent annotation among the five annotations.

| dataset | $r$ | $p$-value |
|---------|--------|-----------|
| MultiNLI | 0.3240 | $\ll 0.01$ |
| SNLI | 0.3350 | $\ll 0.01$ |

Table 4: The *Pearson* correlation coefficients ($r$) between the original PVI estimates and inter-annotator agreement, and the corresponding $p$-values.

Table 4 shows the *Pearson* correlation coefficients between the original PVI estimates and inter-annotator agreement,[4] which reveals a positive correlation between the original PVI estimates and inter-annotator agreement. Although the hardness of an instance for humans is not necessarily equivalent to that for a given model, there should be a positive correlation between them. Thus, the results shown in Table 4 are considered reasonable, which show weak but positive correlations ($r \approx 0.3$) in MultiNLI and SNLI. A weak positive correlation between PVI estimates and inter-annotator agreement can be viewed as a sign of reliability.

Figure 5 shows the *Pearson* correlation coefficients between in-context PVI estimates and inter-annotator agreement. Most of the estimates made by smaller models (with less than 7B parameters) have a negative correlation with inter-annotator agreement. However, even for the smaller models, the correlation becomes more positive as the number of shots increases. A positive correlation between in-context PVI estimates and inter-annotator agreement seems to be associated with model scale, as positive correlations are observed in most of the cases for models larger than 7B.

Table 5 shows the *Pearson* correlation coefficients between the original PVI estimates made by BERT, and between in-context PVI made by GPT3-175B and inter-annotator agreement. It shows that in-context PVI estimates made by this 175B model

---

[4]The original PVI estimates are taken from https://github.com/kawine/dataset_difficulty.

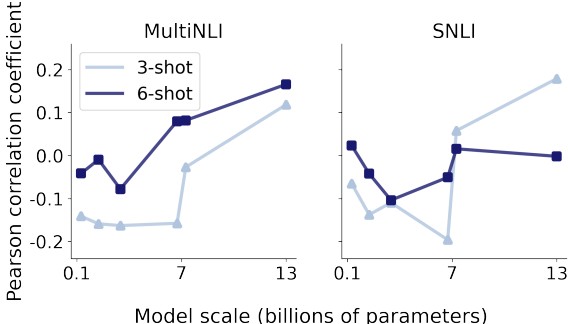

Figure 5: The average *Pearson* correlation coefficients between in-context PVI estimates and inter-annotator agreement for runs on MultiNLI and SNLI using different models and the number of shots. Statistics are averaged over the results obtained using 3 sets of exemplars. Corresponding *p*-values are shown in Table 16 in A.6.

for MultiNLI are very close to the original PVI estimates, despite that they are obtained using only 3-shot prompts. However, in-context PVI estimates made by GPT3-175B for SNLI are less correlated with inter-annotator agreement compared to the original PVI.

| dataset | model | $r$ | $p$-value |
|---------|-------|-----|-----------|
| MultiNLI | BERT | 0.2316 | $\ll 0.01$ |
| | GPT3-175B | 0.2217 | $\ll 0.01$ |
| SNLI | BERT | 0.3678 | $\ll 0.01$ |
| | GPT3-175B | 0.1732 | $\ll 0.01$ |

Table 5: The *Pearson* correlation coefficients ($r$) between the original PVI estimates (made by BERT) and inter-annotator agreement, and between in-context PVI (made by GPT3-175B) and inter-annotator agreement. Results are based on the first 500 test instances in MultiNLI and SNLI. Statistics regarding GPT3-175B are calculated using in-context PVI estimates obtained using 3-shot prompts and are averaged over the results obtained using 3 sets of exemplars.

In a nutshell, in-context PVI estimates made by larger models and larger numbers of shots are more reliable in the sense that they tend to have a more positive correlation with inter-annotator agreement.

### 5.5 In-context PVI for challenging instances

To investigate the potential of in-context PVI in identifying challenging instances, we performed a qualitative analysis of Health Advice and Causal Language. A.1 shows more details of the two tasks.

The results indicate that the most challenging annotations often fall into the class of "no advice" (Health Advice) and "no relationship" (Causal Language). This is primarily due to the confusion

created by some linguistic cues. Certain sentences may contain noticeable advice indicators (such as "should be further explored," as exemplified in the first example of Table 6) which aim at suggesting the need for subsequent investigations. However, based on the annotation schema, sole suggestions pertaining to health-related practices or policies are considered as advice, whereas suggestions for further studies do not fall within the category of it. In addition, according to the annotation schema, a claim that suggests the advantages or effectiveness of an intervention is categorized as "weak advice" or "strong advice". However, terms like "effective" also appear when reporting experimental results, and experimental results are not considered to be advice according to the annotation schema of Health Advice, which makes the instance challenging for the model, such as the second example in Table 6. In Causal Language, some instances contain causal markers such as "association" in subordinate clauses, which do not suggest any relationship according to the annotation schema and thus cause confusion. This is evident from the third and fourth examples in Table 6. These patterns identified through in-context PVI in Health Advice and Causal Language align with the typical prediction error patterns observed in previous studies (Yu et al., 2019; Li et al., 2021). See also Tables 17, 18, and 19 in A.7 for the most challenging instances in each category in MultiNLI and SNLI identified by in-context PVI estimates made by GPT3-175B.

### 5.6 Exemplar selection using in-context PVI

We conducted preliminary experiments to evaluate the effectiveness of in-context PVI in selecting exemplars that enhance performance. Our approach is simple and straightforward: based on in-context PVI estimates obtained using one set of randomly selected exemplars, we selected the most challenging training instance from each category in CoLA, MultiNLI, and SNLI as the exemplars. Intuitively, hard instances are better exemplars, since a model is already proficient in dealing with easier examples, which makes easier examples less valuable for a demonstration. The exemplars we used are shown in Tables 17, 18, and 19 in A.7.

Table 7 indicates that models utilizing the hardest instances as exemplars perform slightly better than those using randomly selected ones for MultiNLI and SNLI. However, this approach leads to a decrease in performance for CoLA. We specu-

| example | target | PVI |
|---|---|---|
| Supplementation with L. reuteri 6475 **should be further explored** as a novel approach to prevent age-associated bone loss and osteoporosis. | no advice | -3.6240 |
| In patients with active ophthalmopathy, teprotumumab **was more effective** than placebo in reducing proptosis and the Clinical Activity Score. | no advice | -3.4560 |
| This work has important policy implications for public health, given the continuous nature of the BMI-IHD **association** and the modifiable nature of BMI. | no relationship | -5.1582 |
| The observed **associations** between pre-diagnostic serum GGT and different breast cancer subtypes may indicate distinct underlying pathways and require further investigations to tease out their clinical implications. | no relationship | -4.9277 |

Table 6: Examples of challenging instances in Health Advice and Causal Language identified by in-context PVI. Expressions that may lead to confusion are **bolded** and marked in red.

| dataset | random | hardest |
|---|---|---|
| CoLA | **0.7571** | 0.7230 |
| MultiNLI | 0.7180 | **0.7380** |
| SNLI | 0.6700 | **0.6980** |

Table 7: The accuracy of GPT3-175B using randomly selected exemplars (random) and the hardest training instances (hardest) as the exemplars. MultiNLI and SNLI results are based on the first 500 testing instances.

late that this is due to some exemplars being mislabeled, which can be misleading to the model. Table 8 shows two of the most challenging instances in CoLA, which are used as the exemplars. At least for the authors of this paper, the first example is not linguistically acceptable, while the second is acceptable.

| sentence | label | PVI |
|---|---|---|
| The harder it has rained, how much faster a flow appears in the river? | acceptable | -13.37 |
| John wrote books. | unacceptable | -11.26 |

Table 8: The hardest instance in each category in CoLA, determined by in-context PVI estimates obtained using GPT3-175B. Examples in red are considered to be mislabeled.

Our findings demonstrate that hard instances can indeed enhance model performance in some cases. However, to fully leverage the power of in-context PVI for selecting exemplars that enhance model performance, it is imperative to adopt a more sophisticated approach. Moreover, we identified challenging instances based on in-context PVI obtained using only one set of randomly selected examplars. Future work should take into account prompt variations.

## 6 Conclusion

This paper introduces in-context PVI, an alternative approach to access the hardness of an instance.

Compared to the original PVI, our proposed method significantly reduces computational cost while behaving similarly to it. We showed that in-context PVI estimates are consistent across different models, selections of exemplars, and numbers of shots. We discovered that larger models tend to produce more reliable in-context PVI estimates, suggesting that in-context PVI is not a strict replacement for the original PVI, and that for smaller models (especially for those that are readily "fine-tunable"), the original PVI is a better choice. We demonstrated that in-context PVI helps identify challenging instances. We also presented a preliminary analysis on the potential of in-context PVI to enhance model performance through exemplar selection.

The utility of in-context PVI for discerning instance difficulty may be effectively harnessed for dataset selection in joint learning scenarios, facilitating the construction of balanced and optimized training sets. Concurrently, future work may utilize in-context PVI as a tool to design an efficient curriculum learning protocol for LLMs (Bengio et al., 2009; Hacohen and Weinshall, 2019). Developing an adaptive curriculum for ICL, progressing from easier to more difficult instances, as gauged by their in-context PVI estimates, represents a significant direction. A further avenue of exploration lies in developing an efficient training method that strategically prioritizes challenging instances, as identified by in-context PVI, thereby creating focused mini-batches for training LLMs. This approach could potentially provide both computational savings and enhance learning efficacy. Achieving these objectives, however, necessitates addressing a series of research questions, including optimizing joint learning dataset composition and creating algorithms that effectively prioritize difficult instances in the training process. These questions constitute the main focus of our future research.

## Limitations

A theoretical analysis is still needed to enhance our understanding and validation of in-context PVI. In order to manage costs, we limited our experiments with OpenAI text-davinci-003 (GPT3-175B) to CoLA, RTE, and the initial 500 test instances in MultiNLI and SNLI. Due to the closed source nature of GPT3-175B, the reproducibility of the results related to it may be a concern as well.

## Acknowledgements

The authors would like to thank the anonymous reviewers for feedback that improved the paper, the funding provided by the LOEWE Distinguished Chair "Ubiquitous Knowledge Processing" (LOEWE initiative, Hesse, Germany), US National Institutes of Health (NIH) grant 5R01GM11435, and the Woods Foundation. Any opinions, findings, conclusions, or recommendations expressed in this material are those of the authors and do not necessarily reflect the views of NIH or the Woods Foundation.

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

# A Appendix

## A.1 Experiment details

All experiments were run using the same configuration on either a NVIDIA A100 or a GTX3090, or through OpenAI API. Tables 9, 10, and 11 give more details of the datasets we used.

| dataset | domain | testing set size | type of task | in Ethayarajh et al. (2022) |
|---|---|---|---|---|
| BoolQ | General | 3270 | Yes/no question and answering | No |
| CoLA | General | 527 | Acceptability | Yes |
| MultiNLI | General | 10000 | Natural language inference | Yes |
| RTE | General | 277 | Entailment | No |
| SNLI | General | 10000 | Natural language inference | Yes |
| Health Advice | Biomedical | 4784 | Suggestion mining | No |
| Causal Language | Biomedical | 2446 | Causal relation | No |

Table 9: Overview of datasets used in this study. These datasets cover a variety of tasks and domains, providing a comprehensive base for our analyses.

| dataset | instance | label |
|---|---|---|
| BoolQ | PASSAGE: "Windows Movie Maker – Windows Movie Maker (formerly known as Windows Live Movie Maker in Windows 7) is a discontinued video editing software by Microsoft. It is a part of Windows Essentials software suite and offers the ability to create and edit videos as well as to publish them on OneDrive, Facebook, Vimeo, YouTube, and Flickr." QUESTION:"is windows movie maker part of windows essentials" | true |
|  | PASSAGE: "The Golden Compass (film) – In 2011, Philip Pullman remarked at the British Humanist Association annual conference that due to the first film's disappointing sales in the United States, there would not be any sequels made." QUESTION:"is there a sequel to the movie the golden compass" | false |
| CoLA | The sailors rode the breeze clear of the rocks. | acceptable |
|  | The more does Bill smoke, the more Susan hates him. | unacceptable |
| MultiNLI | PREMISE: Your gift is appreciated by each and every student who will benefit from your generosity. HYPOTHESIS: Hundreds of students will benefit from your generosity. | neutral |
|  | PREMISE: yes now you know if everybody like in August when everybody's on vacation or something we can dress a little more casual. HYPOTHESIS: August is a black out month for vacations in the company. | contradiction |
|  | PREMISE: At the other end of Pennsylvania Avenue, people began to line up for a White House tour. HYPOTHESIS: People formed a line at the end of Pennsylvania Avenue. | entailment |
| SNLI | PREMISE: This church choir sings to the masses as they sing joyous songs from the book at a church. HYPOTHESIS: The church has cracks in the ceiling. | neutral |
|  | PREMISE: A statue at a museum that no seems to be looking at. HYPOTHESIS: Tons of people are gathered around the statue. | contradiction |
|  | PREMISE: A woman with a green headscarf, blue shirt and a very big grin. HYPOTHESIS: The woman is very happy. | entailment |
| RTE | PREMISE: Valero Energy Corp., on Monday, said it found "extensive" additional damage at its 250,000-barrel-per-day Port Arthur refinery. HYPOTHESIS: Valero Energy Corp. produces 250,000 barrels per day. | entailment |
|  | PREMISE: Oil prices fall back as Yukos oil threat lifted. HYPOTHESIS: Oil prices rise. | no entailment |

Table 10: Instance examples of BoolQ, CoLA, MultiNLI, RTE, SNLI, Health Advice, and Causal Language.

| dataset | instance | label |
|---|---|---|
| Health Advice | Nurses should assess patient decision-making styles to ensure maximum patient involvement in the decision-making process based on personal desires regardless of age. | strong advice |
| | Adolescents with high risk factors, especially those with menstrual disorders and hyperandrogenism, may need careful clinical screening | weak advice |
| | Former smokers are at risk for hypertension, probably because of the higher prevalence of overweight and obese subjects in this group. | no advice |
| Causal Langauge | The findings from this large prospective study show that men who are taller and who have greater adiposity have an elevated risk of high-grade prostate cancer and prostate cancer death. | correlational |
| | MTHFR A1298C polymorphism might contribute to an increased risk of breast cancer and/or ovarian cancer susceptibility. | conditional causal |
| | Participatory community-based nutrition education for caregivers improved child dietary diversity even in a food insecure area. | direct causal |
| | This approach may, however, be difficult to implement on a large scale. | no relation |

Table 11: Continuation of Table 10.

## A.2 Example of prompts

Table 12 and Table 13 are examples of the prompts we used. Our prompt design can be cross-validated with Chen et al. (2023).

---

CONTEXT: How much harder has it rained, the faster a flow you see in the river?
QUESTION: Is this (0) unacceptable, or (1) acceptable?
ANSWER: 1

CONTEXT: The more obnoxious Fred, the less attention you should pay to him.
QUESTION: Is this (0) unacceptable, or (1) acceptable?
ANSWER: 0

CONTEXT: I'm glad I saw anybody.
QUESTION: Is this (0) unacceptable, or (1) acceptable?
ANSWER: 0

CONTEXT: Julie and Jenny arrived first
QUESTION: Is this (0) unacceptable, or (1) acceptable?
ANSWER: 1

---

Table 12: An example of a 4-shot *input-target* prompt for CoLA.

---

ANSWER: 1

ANSWER: 0

ANSWER: 0

ANSWER: 1

---

Table 13: An example of a 4-shot *null-target* prompt for CoLA.

## A.3 More on general results

Table 14 shows a broad overview of in-context PVI estimates across datasets and models, along with their relationship with prediction accuracy.

| dataset | accuracy | acc. low PVI | acc. high PVI | PVI for TRUE | PVI for FALSE |
|---|---|---|---|---|---|
| BoolQ | 0.4543 | 0.3173 | 0.6241 | 0.2009 | -0.1075 |
| CoLA | 0.5175 | 0.3374 | 0.6586 | 0.2416 | -0.1486 |
| MultiNLI | 0.3483 | 0.2126 | 0.4691 | 0.6609 | 0.1094 |
| RTE | 0.4482 | 0.3330 | 0.5287 | 0.8452 | 0.7243 |
| SNLI | 0.3426 | 0.2715 | 0.3929 | 0.5872 | 0.2482 |
| Health Advice | 0.3027 | 0.1874 | 0.4336 | 0.4162 | 0.0101 |
| Causal Language | 0.2277 | 0.1321 | 0.3499 | 0.5136 | -0.1918 |

(a) dataset-wise

| model | accuracy | acc. low PVI | acc. high PVI | PVI for TRUE | PVI for FALSE |
|---|---|---|---|---|---|
| GPT2-125M | 0.3753 | 0.2077 | 0.5523 | 0.7691 | 0.2888 |
| GPT-Neo-1.3B | 0.3610 | 0.3077 | 0.3930 | 0.1842 | 0.0808 |
| GPT-Neo-2.7B | 0.3672 | 0.3141 | 0.4137 | 0.3880 | 0.2858 |
| GPT-J-6B | 0.3775 | 0.3553 | 0.4199 | 0.1292 | 0.0823 |
| GPT-JT-6B | 0.3811 | 0.2792 | 0.4555 | 0.2156 | 0.0129 |
| GPT3-175B* | 0.7167 | 0.0104 | 0.9763 | 2.0003 | -5.0125 |
| Alpaca-7B | 0.3791 | 0.1414 | 0.6018 | 0.6852 | -0.2365 |
| Alpaca-13B | 0.3975 | 0.1847 | 0.6077 | 1.4344 | 0.4778 |

(b) model-wise

Table 14: The average prediction accuracy, the average prediction accuracy for instances with the lowest 20% in-context PVI estimates (acc. low PVI), the average prediction accuracy for instances with the highest 20% in-context PVI estimates (acc. high PVI), average in-context PVI estimates for correct predictions (PVI for TRUE), and average in-context PVI estimates for incorrect predictions (PVI for FALSE) for runs using different datasets or models. Statistics are averaged over the results obtained using 3 sets of exemplars, and 2 numbers of shots. GPT3-175B was only tested on CoLA, RTE, and the first 500 test instances in MultiNLI and SNLI.

## A.4 More on the consistency of in-context PVI

Figure 6 shows the *Pearson* correlation coefficients between in-context PVI estimates across different models.

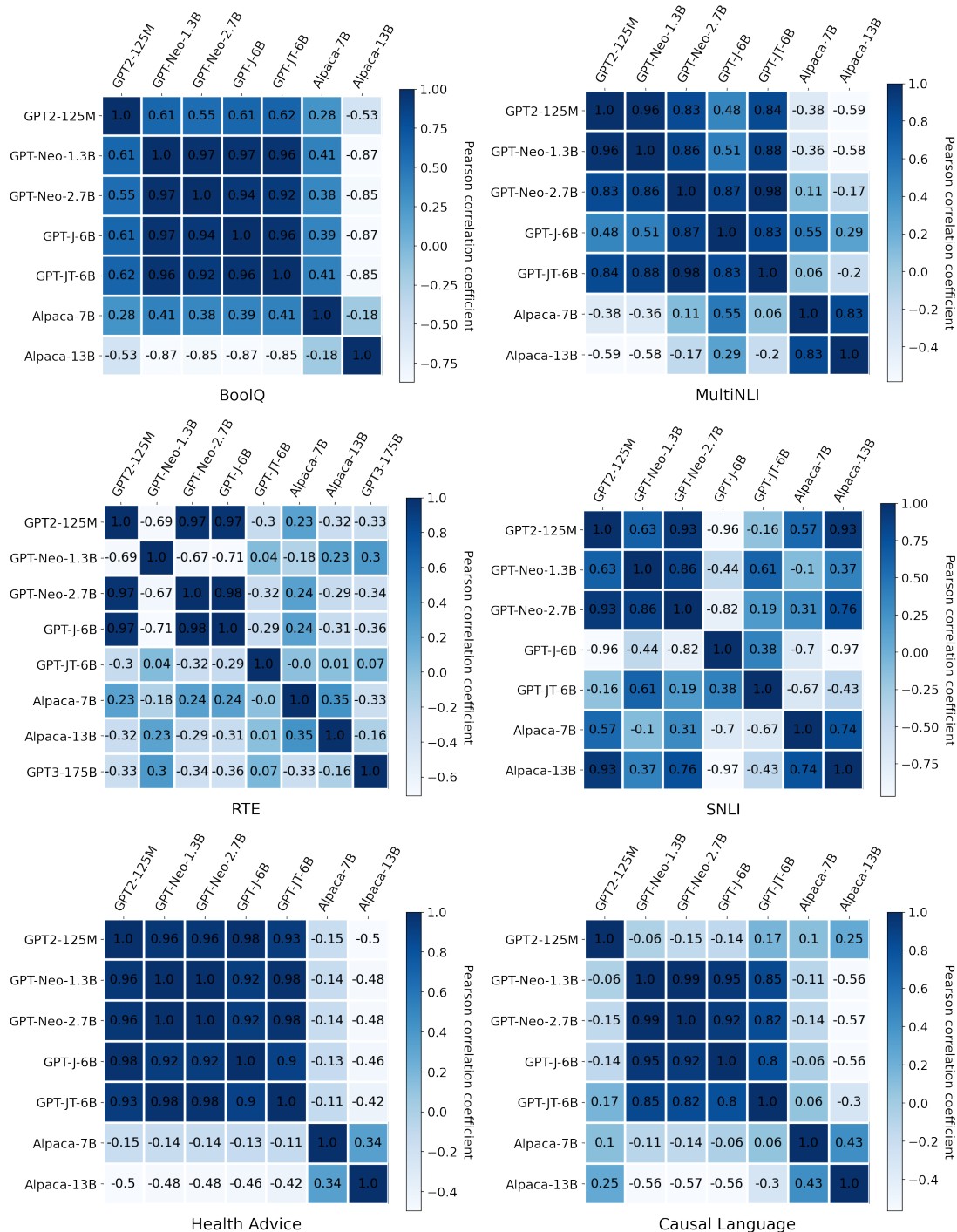

Figure 6: The *Pearson* correlation coefficients among in-context PVI estimates made by different models. In-context PVI estimates are obtained using prompts that contain the minimal number of exemplars, i.e., the number of unique labels in each dataset. CoLA is an exception. The minimal number of exemplars for CoLA is 4, since it contains only 2 labels and the input sentences are short.

## A.5 More on the variance of in-context PVI

Table 15 shows the full results of one-way ANOVA for the variance of in-context PVI estimates across different sets of exemplars.

| dataset | model | F-statistic | p-value |
|---|---|---|---|
| BoolQ | GPT2-125M | 1.0405 | 0.4537 |
| | GPT-Neo-1.3B | 0.2433 | 0.7981 |
| | GPT-Neo-2.7B | 0.5357 | 0.6325 |
| | GPT-J-6B | 0.1484 | 0.8680 |
| | GPT-JT-6B | 0.1978 | 0.8305 |
| | Alpaca-7B | 0.9711 | 0.4729 |
| | Alpaca-13B | 0.3944 | 0.7046 |
| CoLA | GPT2-125M | 0.3024 | 0.7592 |
| | GPT-Neo-1.3B | 0.8949 | 0.4957 |
| | GPT-Neo-2.7B | 0.1543 | 0.8634 |
| | GPT-J-6B | 2.3202 | 0.2460 |
| | GPT-JT-6B | 0.1690 | 0.8520 |
| | Alpaca-7B | 1.8573 | 0.2986 |
| | Alpaca-13B | 1.0781 | 0.4438 |
| MultiNLI | GPT2-125M | 0.2958 | 0.7634 |
| | GPT-Neo-1.3B | 0.5543 | 0.6239 |
| | GPT-Neo-2.7B | 0.0383 | 0.9629 |
| | GPT-J-6B | 0.2604 | 0.7866 |
| | GPT-JT-6B | 0.3099 | 0.7545 |
| | Alpaca-7B | 0.0363 | 0.9647 |
| | Alpaca-13B | 0.4304 | 0.6850 |
| RTE | GPT2-125M | 0.4876 | 0.6556 |
| | GPT-Neo-1.3B | 0.0803 | 0.9247 |
| | GPT-Neo-2.7B | 0.7250 | 0.5535 |
| | GPT-J-6B | 1.2598 | 0.4007 |
| | GPT-JT-6B | 1.5383 | 0.3468 |
| | Alpaca-7B | 0.4289 | 0.6857 |
| | Alpaca-13B | 0.2346 | 0.8042 |
| SNLI | GPT2-125M | 0.3518 | 0.7290 |
| | GPT-Neo-1.3B | 1.5983 | 0.3369 |
| | GPT-Neo-2.7B | 3.0431 | 0.1897 |
| | GPT-J-6B | 0.7946 | 0.5285 |
| | GPT-JT-6B | 62.8332 | **0.0036** |
| | Alpaca-7B | 0.1426 | 0.8726 |
| | Alpaca-13B | 0.0905 | 0.9158 |
| Health Advice | GPT2-125M | 0.3786 | 0.7135 |
| | GPT-Neo-1.3B | 5.7712 | 0.0937 |
| | GPT-Neo-2.7B | 0.0386 | 0.9626 |
| | GPT-J-6B | 1.3833 | 0.3752 |
| | GPT-JT-6B | 0.4463 | 0.6766 |
| | Alpaca-7B | 0.1852 | 0.8398 |
| | Alpaca-13B | 0.6809 | 0.5704 |
| Causal Language | GPT2-125M | 0.1922 | 0.8345 |
| | GPT-Neo-1.3B | 36.8063 | **0.0077** |
| | GPT-Neo-2.7B | 12.0928 | **0.0367** |
| | GPT-J-6B | 0.4149 | 0.6933 |
| | GPT-JT-6B | 1.6823 | 0.3236 |
| | Alpaca-7B | 0.0084 | 0.9917 |
| | Alpaca-13B | 1.3805 | 0.3758 |

Table 15: The results of one-way ANOVA for the variance of in-context PVI estimates across different sets of exemplars. p-values that are smaller than 0.05 are **bolded**, which suggest that there are significant differences in in-context PVI estimates obtained using different sets of exemplars.

## A.6 More on the correlation between in-context PVI and inter-annotator agreement

| dataset | model | number of shots | $r$ | $p$-value |
|---|---|---|---|---|
| MultiNLI | GPT2-125M | 3 | -0.1412 | 4.61E-34 |
| | | 6 | -0.0416 | 2.83E-01 |
| | GPT-Neo-1.3B | 3 | -0.1590 | 2.44E-46 |
| | | 6 | -0.0096 | 1.22E-01 |
| | GPT-Neo-2.7B | 3 | -0.1630 | 7.18E-46 |
| | | 6 | -0.0780 | 1.08E-18 |
| | GPT-J-6B | 3 | -0.1782 | 9.74E-68 |
| | | 6 | -0.1725 | 2.26E-59 |
| | GPT-JT-6B | 3 | -0.1577 | 7.62E-41 |
| | | 6 | 0.0798 | 3.56E-13 |
| | Alpaca-7B | 3 | -0.0271 | 5.28E-04 |
| | | 6 | 0.0817 | 7.20E-02 |
| | Alpaca-13B | 3 | 0.1183 | 6.87E-08 |
| | | 6 | 0.1656 | 6.27E-23 |
| SNLI | GPT2-125M | 3 | -0.0656 | 2.68E-48 |
| | | 6 | 0.0230 | 2.36E-02 |
| | GPT-Neo-1.3B | 3 | -0.1376 | 2.19E-03 |
| | | 6 | -0.0423 | 1.82E-42 |
| | GPT-Neo-2.7B | 3 | -0.1097 | 1.49E-17 |
| | | 6 | -0.1039 | 2.80E-11 |
| | GPT-J-6B | 3 | -0.2042 | 7.14E-77 |
| | | 6 | -0.1116 | 4.34E-18 |
| | GPT-JT-6B | 3 | -0.1961 | 1.29E-73 |
| | | 6 | -0.0508 | 3.90E-49 |
| | Alpaca-7B | 3 | 0.0578 | 8.66E-05 |
| | | 6 | 0.0155 | 6.06E-02 |
| | Alpaca-13B | 3 | 0.1787 | 1.07E-21 |
| | | 6 | -0.0019 | 4.67E-05 |

Table 16: The average *Pearson* correlation coefficients ($r$) and corresponding $p$-values between in-context PVI estimates and inter-annotator agreement for runs on MultiNLI and SNLI using different models and the number of shots. Statistics are averaged over the results obtained using 3 sets of exemplars.

## A.7 More on challenging instances

| sentence | target | PVI |
|---|---|---|
| The harder it has rained, how much faster a flow appears in the river? | acceptable | -13.3701 |
| As John eats more, keep your mouth shut tighter, OK? | acceptable | -13.0345 |
| The more people you say that will buy tickets, the happier I'll be. | unacceptable | -11.4478 |
| John wrote books. | unacceptable | -11.2615 |

Table 17: The hardest instances in each category in CoLA, determined by in-context PVI estimates obtained using GPT3-175B.

| sentences | target | PVI |
|---|---|---|
| SENTENCE1: It's not that the questions they asked weren't interesting or legitimate (though most did fall under the category of already asked and answered). SENTENCE2: All of the questions were interesting according to a focus group consulted on the subject. | neutral | -12.0748 |
| SENTENCE1: i know i am um i don't know anybody in their right mind that says that that i'm doing it because i want to i SENTENCE2: I know there are people who think I'm doing it because I desire to. | contradiction | -8.1535 |
| SENTENCE1: And in another shift in the economy, it was found that lamb could be raised more cost-effectively on lowland farms in part because of the richer, more nutritious grazing land available there and as a result Lakeland farms became less profitable. SENTENCE2: Another shift in the economy was found to be more nutritious. | entailment | -6.7892 |

Table 18: The hardest instance in each category in the first 500 training instances in MultiNLI, determined by in-context PVI estimates obtained using GPT3-175B.

| sentences | target | PVI |
|---|---|---|
| SENTENCE1: A white horse is pulling a cart while a man stands and watches. 
 SENTENCE2: A horse is hauling goods. | neutral | -9.8986 |
| SENTENCE1: A couple holding hands walks down a street. 
 SENTENCE2: There are people sitting on the side of the road. | contradiction | -8.3236 |
| SENTENCE1: A foreign family is walking along a dirt path next to the water. 
 SENTENCE2: A foreign family walks by a dirt trail along a body of water. | entailment | -1.4902 |

Table 19: The hardest instance in each category in the first 500 training instances in SNLI, determined by in-context PVI estimates obtained using GPT3-175B.

