# OpenReview forum: "Measuring Pointwise $\mathcal{V}$-Usable Information In-Context-ly"
_EMNLP/2023/Conference — EMNLP 2023 Findings_

### Official Review · Reviewer_pfMK · 2023-07-25

**Typos Grammar Style And Presentation Improvements:** line 257
**Soundness:** 3

**Excitement:**

3: Ambivalent: It has merits (e.g., it reports state-of-the-art results, the idea is nice), but there are key weaknesses (e.g., it describes incremental work), and it can significantly benefit from another round of revision. However, I won't object to accepting it if my co-reviewers champion it.

**Paper Topic And Main Contributions:**

This paper generalize the method of using pointwise v-usable information to analysis instance hardness w.r.t. fine-tuning to that w.r.t. in-context learning. The authors first showed strong correlations between the proposed metric and the accuracy, justifying that it is a reasonable hardness metric. They next demonstrate the reliability of the newly proposed measure, including relatively consistent results w.r.t. different exemplar sets and shot numbers, and low variance among different exemplar sets. They also use the metric to identify hard instances and use these instances to improve in-context learning.

**Reasons To Accept:**

It is novel to generalize pointwise v-usable information from fine-tuning to in-context learning.

**Reasons To Reject:**

(1) My major concern lies in the accuracy difference between fine-tuning and in-context learning. In [1], fine-tuning achieves 80%~90% accuracy on the tested datasets, while in this paper, Table 1 (the first column) shows that in-context learning's accuracy on all the seven datasets are below or similar to random guess, i.e., in-context learning cannot make any meaningful predictions on these datasets. I think this may make the analysis in this paper invalid, and the authors could test on other datasets where in-context learning has a relatively competent performance.

Specifically, in Eq(1), the first term only depends on the input label and is identical for instances with the same label; the second term reflects the model's prediction. But if this prediction is merely a random number, the hardness measured by Eq(1) is also a random number without any meaning. Because the model's prediction is defined in a similar way as the second term, it is straightforward that the value of Eq(1) will correlate well with the model's accuracy (which the rest of Table 1 shows), but this cannot be taken as a justification of the metric.

(2) It is known that the performance of in-context learning depends on the quality of the in-context exemplars (and the number of shots), thus many methods of selecting good exemplars has been proposed. But in section 5.1 the authors conclude that pointwise v-usable information is stable w.r.t. different exemplar sets. This seems contradictory.



[1] Kawin Ethayarajh, Yejin Choi, and Swabha Swayamdipta. Understanding dataset difficulty with v-usable information. ICML 2022.

**Reproducibility:**

4: Could mostly reproduce the results, but there may be some variation because of sample variance or minor variations in their interpretation of the protocol or method.

**Reviewer Confidence:**

2: Willing to defend my evaluation, but it is fairly likely that I missed some details, didn't understand some central points, or can't be sure about the novelty of the work.

---

> ### Author Rebuttal · Authors · 2023-08-28
>
> Thank you very much for your review.
>
> **Reason-to-Reject #1: My major concern lies in the accuracy difference between fine-tuning and in-context learning. In [1], fine-tuning achieves 80%~90% accuracy on the tested datasets, while in this paper, Table 1 (the first column) shows that in-context learning's accuracy on all the seven datasets are below or similar to random guess, i.e., in-context learning cannot make any meaningful predictions on these datasets. I think this may make the analysis in this paper invalid, and the authors could test on other datasets where in-context learning has a relatively competent performance.**
>
> We would like to clarify that the results in Table 1 are the averaged results over 7 models, 3 sets of exemplars, and 2 numbers of shots. There is a large difference between the performance of smaller models and larger models. The following table shows the accuracy and in-context PVI related results of GPT2-125M and GPT3-175B on RTE using a 2-shot prompt:
>
> |model|accuracy|accuracy for instances with the lowest 20% PVI|accuracy for instances with the highest 20% PVI|
> |--|--|--|--|
> |GPT2-125M|42.4|20.8|59.3|
> |GPT3-175B|74.0|4.2|93.5|
>
>  As a reference, the accuracy of fine-tuned BERT-large on RTE is 70.1%, and the accuracy of fine-tuned RoBERTa is 86.6% [1, 2]. GPT3-175B has a solid performance compared to the two fine-tuned models.
>
> Table 1 gives a broad overview of in-context PVI estimates across datasets. What we are trying to convey in Table 1 is that the prediction accuracy differs for instances with low and high in-context PVI estimates, and that in-context PVI estimates for correctly predicted instances are higher than those for incorrectly predicted instances, which align with what we would logically expect for a hardness metric.
>
> We will replace Table 1 with the results of the best performing model (i.e., GPT3-175B), and we will also report specific in-context PVI results of each model in the appendix (please also see our reply to Reviewer1, Reason-to-Reject #1).
>
> [1] Devlin, J., Chang, M. W., Lee, K., & Toutanova, K. (2018). Bert: Pre-training of deep bidirectional transformers for language understanding. arXiv preprint arXiv:1810.04805.
> [2] Liu, Y., Ott, M., Goyal, N., Du, J., Joshi, M., Chen, D., Levy, O., Lewis, M., Zettlemoyer, L. & Stoyanov, V. (2019). Roberta: A robustly optimized bert pretraining approach. arXiv preprint arXiv:1907.11692.
>
> **Reason-to-Reject #2: It is known that the performance of in-context learning depends on the quality of the in-context examples (and the number of shots), thus many methods of selecting good examples has been proposed. But in section 5.1 the authors conclude that pointwise v-usable information is stable w.r.t. different exemplar sets. This seems contradictory.**
>
> Previous work has shown that the performance of ICL depends on the selection of exemplars, and we examine this in our paper, where we perform a comprehensive analysis regarding the variance of in-context PVI estimates across different sets and different numbers of randomly selected exemplars. As reported in section 5.2, our results show that in-context PVI estimates are quite stable. Our results show that the stability of in-context PVI estimates is not as sensitive to the selection of exemplars as the accuracy of in-context learners.
>
> For more information regarding our method to select exemplars and numbers of shots, please refer to our response to Reviewer1 Question-B and Reviewer2 Question-A.
>
> **Typos Grammar Style And Presentation Improvements: line 257: should it be "three unique labels"?**
>
> Thank you for noticing this. We have modified it.

---

### Official Review · Reviewer_Ydp3 · 2023-08-03

**Soundness:** 3

**Excitement:**

4: Strong: This paper deepens the understanding of some phenomenon or lowers the barriers to an existing research direction.

**Paper Topic And Main Contributions:**

This paper proposes a new way to measure pointwise $\mathcal{V}$-usable information using in-context learning. Instead of fine-tuning models on input-output and output data, their approach prompts the models with demonstrations. The authors empirically assess the robustness of in-context PVI measure to the variations in the prompt, and find that the in-context PVI is quite robust. Furthermore, they measure the correlation between inter annotator agreement and in-context PCI and find that larger models correlate with inter-annotator agreement better. Finally, they employ in-context PVI as a way to select demonstrations for in-context learning.

**Questions For The Authors:**

- Table 1: How demonstrations and the number of demonstrations are chosen for these results? Is it a random choice of demonstrations (and number of demonstrations) or are they chosen based on how well they correlated with accuracy?
- The caption and explanations for Table 5 are confusing. I think you are only measuring the correlation between in-context PVI and inter-annotator agreement. But the caption suggests that you are also measuring the correlation with the original-PVI, which one is that?
- I can not fully understand section 5.5. What confuses me is that the task is not explained upfront. So it is hard to understand why some samples are "challenging".
- In lines 381-384 you explain why the hardness of an instance for humans is not necessarily equivalent to that for a given model. I very much agree with this point. What would be particularly interesting is to look into the samples that are hard for the model and not hard for humans (or visa versa), Do you have any insights on this? Do you think in-context PVI can help uncover datasets artifacts through such analysis?

**Reasons To Accept:**

- Computing PVI for large language models can be expensive. Particularly, for models with API access (like GPT-3) it is infeasible to fine-tune models to compute PVI. The method introduced in this paper does not require fine-tuning models and aims to measure PVI only through prompting.
- Based on the empirical results, in-context PVI seems to correlate positively with accuracy, correlate with inter-annotator agreement, which suggests that it can serve as a replacement for PVI.

**Reasons To Reject:**

- I believe whether or not to use in-context PVI or PVI would depend on how we want to use the model. If the model is going to be prompted to solve the downstream task, it makes more sense to use in-context PVI. However, especially in scenarios where in-context performance is far from the state-of-the-art fine-tuned model, it would be best to use PVI to measure dataset difficulty. In that scenario, I think the paper lacks a more thorough and theoretical discussion on how PVI and in-context PVI relate to each other (e.g., Is one a lower bound for another?)
- It seems necessary to compare and contrast in-context PVI with PVI directly. I am not convinced why PVI and in-context PVI are only measured in comparison (w.t.r. correlation) to inter-annotator but the correlation of the two measures is not assessed directly.
- Significance of the results for Table 1., Figure 5., and Table 7 seems to be necessary and is not reported. For instance, in line 268, the authors claim that low in-context PVI estimates are **significantly** lower than that for instance with high in-context PVI estimates. The corresponding results in Table 1, however, do not report any statistical significance level, and thus do not support this claim. My suggestion would be to directly measure the correlation between in-context PVI and accuracy and report statistical significance.
- After reading the last experiment, I am not convinced that in-context PVI should be used for selecting demonstrations. The reason is that the performance gain in Table 7 is not quite large and again the significance is not measured. Particularly, due to the large body of research showing the variance of performance with prompt variations, it seems necessary to control for this in such experiments. My suggestion is to vary the demonstrations and report the average and standard dev. of accuracy, maybe also doing statistical tests to prove the significance of the result.

**Reproducibility:**

4: Could mostly reproduce the results, but there may be some variation because of sample variance or minor variations in their interpretation of the protocol or method.

**Reviewer Confidence:**

4: Quite sure. I tried to check the important points carefully. It's unlikely, though conceivable, that I missed something that should affect my ratings.

---

> ### Author Rebuttal · Authors · 2023-08-28
>
> Thank you very much for your review.
>
> **Reason-to-Reject #1: I believe whether or not to use in-context PVI or PVI would depend on how we want to use the model. If the model is going to be prompted to solve the downstream task, it makes more sense to use in-context PVI. However, especially in scenarios where in-context performance is far from the state-of-the-art fine-tuned model, it would be best to use PVI to measure dataset difficulty. In that scenario, I think the paper lacks a more thorough and theoretical discussion on how PVI and in-context PVI relate to each other (e.g., Is one a lower bound for another?)**
>
> We very much agree that whether or not to use in-context PVI depends on how we want to use the model. We agree that if fine-tuning leads to a better performance, and it is not prohibitively costly to fine-tune the model, we should use the original PVI. We do not propose in-context PVI as a replacement for the original PVI in this case. In fact, we show in Section 5.3 and 5.4 that in-context PVI estimates made by smaller models (which are more readily “fine-tunable”) have lower qualities, which implies that the original PVI is a better choice for those models. We believe that the primary value of in-context PVI is for cases where fine-tuning is challenging, such as for very large models like GPT3-175B. In the revised version of the paper, we will make it clearer that in-context PVI is not meant to be a strict replacement for the original PVI in cases where fine-tuning is possible (i.e., for smaller models, the original PVI is better than in-context PVI).
>
> **Reason-to-Reject #2: It seems necessary to compare and contrast in-context PVI with PVI directly. I am not convinced why PVI and in-context PVI are only measured in comparison (w.t.r. correlation) to inter-annotator but the correlation of the two measures is not assessed directly.**
>
> We agree that it is more intuitive to directly compare in-context PVI with the original PVI (such as correlating in-context PVI estimates with the original PVI estimates). We did not do that for the following reasons:
>
> * It would be prohibitively costly to obtain original PVI estimates for larger models. The calculation of the original PVI requires fine-tuning. Though it is feasible to fine-tune a model such as GPT2-125M, fine-tuning larger models such as GPT3-175B requires much more computational resources.
> * Our experimental results have shown that in-context PVI estimates from smaller models have lower quality than those from larger models.
>   * We show in Section 5.3 that in-context PVI estimates from GPT2-125M are much noisier than those from GPT3-175B. This suggests that in-context PVI estimates from larger models better capture information useful to make correct predictions.
>   * We show in Section 5.4 that in-context PVI estimates from smaller models are less reliable because they are negatively correlated with the inter-annotator agreement score. Results also show that in-context PVI estimates from larger models such as GPT3-175B are positively correlated with the inter-annotator agreement score, which indicates high reliability.
>
> To summarize, in-context PVI estimates made by larger models are more reliable and they better capture information useful for making correct predictions. However, it is prohibitively costly to obtain their original PVI estimates. While it is feasible to obtain original PVI estimates by fine-tuning smaller models, our evidence has shown that in-context PVI estimates made by smaller models are less reliable. Thus, we do not include a section in our paper that directly compares in-context PVI estimates with the original PVI. Instead, we provide an indirect comparison in section 5.4.
>
> **Reason-to-Reject #3: Significance of the results for Table 1., Figure 5., and Table 7 seems to be necessary and is not reported. For instance, in line 268, the authors claim that low in-context PVI estimates are significantly lower than that for instance with high in-context PVI estimates. The corresponding results in Table 1, however, do not report any statistical significance level, and thus do not support this claim. My suggestion would be to directly measure the correlation between in-context PVI and accuracy and report statistical significance.**
>
> For Table 1, we did not base our claim that “the prediction accuracy for instances with low in-context PVI estimates is significantly lower than that for instances with high in-context PVI estimates” on statistical analysis such as Pearson correlation. Our claim is simply based on the observation that, for example, on BoolQ, the accuracy for instances with low in-context PVI is 0.3173, and the accuracy for instances with high in-context PVI is 0.6241, between which we think there is a big gap (we used the word "significant" to describe it). We have revised the texts that describe Table 1 and 7 to disambiguate.
>
> We confirm that 23 out of 28 results are statistically significant in Figure 5. We will report the significance values in the revised paper.
>
> **Reason-to-Reject #4: After reading the last experiment, I am not convinced that in-context PVI should be used for selecting demonstrations. The reason is that the performance gain in Table 7 is not quite large and again the significance is not measured. Particularly, due to the large body of research showing the variance of performance with prompt variations, it seems necessary to control for this in such experiments. My suggestion is to vary the demonstrations and report the average and standard dev. of accuracy, maybe also doing statistical tests to prove the significance of the result.**
>
> Indeed, the performance gain shown in Table 7 in Section 5.6 is not large. As we pointed out in Section 5.6, we took a straightforward approach, simply selecting the hardest training instances as exemplars. Our approach is based on the intuition that hard instances are better exemplars. We also pointed out that this straightforward approach only helps improve model performance in MultiNLI and SNLI, and we advocated for a more sophisticated approach. This is a potential research question, and we agree that future work should take into account prompt variations in the experiments.
>
> **Reviewer2-Question-A: Table 1: How demonstrations and the number of demonstrations are chosen for these results? Is it a random choice of demonstrations (and number of demonstrations) or are they chosen based on how well they correlated with accuracy?**
>
> Demonstrations are chosen randomly from the training set of each task, and the number of demonstrations is chosen according to the number of unique labels in a dataset. For instance, MultiNLI has 3 unique labels, so we tried 3- and 6-shot (please refer to Exemplar selection in Section 4 and Table 12 for examples of the demonstrations). We did not choose them based on the performance. Please also see our response to Reviewer1, Question-B regarding exemplar selection.
>
> **Reviewer2-Question-B: The caption and explanations for Table 5 are confusing. I think you are only measuring the correlation between in-context PVI and inter-annotator agreement. But the caption suggests that you are also measuring the correlation with the original-PVI, which one is that?**
>
> Thank you for catching that error. In Table 5, apart from the correlation between in-context PVI and inter-annotator agreement, we also report the correlation between the original PVI estimates and inter-annotator agreement, which is taken from [1]. We have updated the caption and explanations for Table 5.
>
> [1] Ethayarajh, K., Choi, Y., & Swayamdipta, S. (2022, June). Understanding Dataset Difficulty with V-Usable Information. In International Conference on Machine Learning (pp. 5988-6008). PMLR.
>
> **Reviewer2-Question-C: I can not fully understand section 5.5. What confuses me is that the task is not explained upfront. So it is hard to understand why some samples are "challenging".**
>
> We will add the task descriptions for Health Advice and Causal Language to the manuscript, and we will carefully edit Section 5.5 to enhance its clarity.
>
> In general, the challenging examples are caused by the potential discrepancy between common sense and annotation schema. For example, a sentence might contain noticeable advice indicators (such as “should be further explored,” as exemplified in the first example of Table 6) aimed at delineating the implications for subsequent investigations. However, based on the annotation schema, solely suggestions pertaining to health-related behavior or policy changes are labeled as health advice, whereas recommendations for forthcoming studies do not fall within the category of health advice. By utilizing the in-context pvi estimates, we are able to identify these examples.
>
> **Reviewer2-Question-D: In lines 381-384 you explain why the hardness of an instance for humans is not necessarily equivalent to that for a given model. I very much agree with this point. What would be particularly interesting is to look into the samples that are hard for the model and not hard for humans (or visa versa), Do you have any insights on this? Do you think in-context PVI can help uncover datasets artifacts through such analysis?**
>
> In Table 8, we present two of the hardest instances discovered by in-context PVI in CoLA. Although the grammatical acceptability of the sentence “John wrote books” is obvious for English speakers, it is considered to be hard here by the model because it was mislabeled (the gold label for that sentence is UNACCEPTABLE, which is incorrect in our opinion). Table 6 shows challenging instances identified by in-context PVI, such as the sentence "Supplementation with L. reuteri 6475 should be further explored as a novel approach to prevent age-associated bone loss and osteoporosis." According to the annotation schema of Health Advice, this sentence should be labeled as "no advice", as it just calls for the need for further scientific study, rather than gives recommendations for health-related behavior or policy changes. However, a general domain model may find the expression "should be further explored" as indicative of advice, making this instance challenging for it. The annotation errors (Table 8) and potential discrepancy between common sense and annotation schema (Table 6) are examples of dataset artifacts that can be uncovered by in-context PVI.

---

### Official Review · Reviewer_jyth · 2023-08-04

**Soundness:** 3

**Excitement:**

3: Ambivalent: It has merits (e.g., it reports state-of-the-art results, the idea is nice), but there are key weaknesses (e.g., it describes incremental work), and it can significantly benefit from another round of revision. However, I won't object to accepting it if my co-reviewers champion it.

**Paper Topic And Main Contributions:**

This paper introduces in-context pointwise V-usable information (in-context PVI), a modified version of the recently proposed hardness metric PVI, tailored for the ICL context. In-context PVI requires only a few exemplars and eliminates the need for fine-tuning. To validate the reliability of in-context PVI, a comprehensive empirical analysis was conducted. The results demonstrate that in-context PVI estimates exhibit similar characteristics to the original PVI. In the in-context setting, the estimates remain consistent across different exemplar selections and numbers of shots, indicating their stability. The work emphasizes the potential of in-context PVI as a tool for assessing model performance and provides valuable contributions to the understanding and application of ICL in the context of large language models.

**Questions For The Authors:**

A: Are there other natural language descriptions for Prompt? Would it be helpful to include a more detailed task description?

B: The current strategy is to select exemplar randomly, is there any plan to improve?


**Reasons To Accept:**

- This paper introduces in-context PVI, a new approach for calculating PVI that exhibits higher efficiency compared to the original PVI method.
- The paper offers a comprehensive empirical analysis, which serves to establish the reliability of in-context PVI. The findings provide new understanding of the capabilities of In-Context Learning (ICL).


**Reasons To Reject:**

- It is better to put the specific in-context PVI results of each model in the appendix for easy comparison.
- Although the current experiment is relatively comprehensive, it is still limited to the comparison between in-context PVI, and it is recommended to compare them with traditional PVI.


**Reproducibility:**

4: Could mostly reproduce the results, but there may be some variation because of sample variance or minor variations in their interpretation of the protocol or method.

**Reviewer Confidence:**

4: Quite sure. I tried to check the important points carefully. It's unlikely, though conceivable, that I missed something that should affect my ratings.

---

> ### Author Rebuttal · Authors · 2023-08-28
>
> Thank you very much for your review.
>
> **Reason-to-Reject #1: It is better to put the specific in-context PVI results of each model in the appendix for easy comparison.**
>
> We will add the detailed results in the appendix that compare in-context PVI estimates across models. The table below gives an example of the results we will add in the appendix. This table shows the overall accuracy and in-context PVI related results for CoLA from different models.
>
> |model|overall accuracy|accuracy for instances with the lowest 20% in-context PVI|accuracy for instances with the highest 20% in-context PVI|
> |--|--|--|--|
> |GPT-neo-1.3B|46.21|46.53|53.38|
> |GPT-neo-2.7B|48.29|46.86|51.97|
> |GPT-J-6B|50.03|45.13|54.25|
> |Alpaca-7B|52.69|15.57|85.49|
> |Alpaca-13B|56.26|13.68|86.45|
> |GPT3-175B|72.42|0|100|
>
> **Reason-to-Reject #2: Although the current experiment is relatively comprehensive, it is still limited to the comparison between in-context PVI, and it is recommended to compare them with traditional PVI.**
>
> We would like to clarify that in Section 5.4, we reported the comparison between in-context PVI and the original PVI in an indirect way: via the correlation between in-context PVI estimates and inter-annotator agreement, and between the original PVI estimates and inter-annotator agreement (see Table 5). Although we very much agree that it is more intuitive to directly compare in-context PVI with the original PVI (such as correlating in-context PVI estimates with the original PVI estimates), we did not do that for several reasons:
>
> * It would be prohibitively costly to obtain original PVI estimates for larger models. The calculation of the original PVI requires fine-tuning. Though it is feasible to fine-tune a model such as GPT2-125M, fine-tuning larger models such as GPT3-175B requires much more computational resources.
> * Our experimental results have shown that in-context PVI estimates from smaller models have lower qualities than those from larger models.
>   * We show in Section 5.3 that in-context PVI estimates from GPT2-125M are much noisier than those from GPT3-175B. This suggests that in-context PVI estimates from larger models better capture information useful to make correct predictions.
>   * We show in Section 5.4 that in-context PVI estimates from smaller models are less reliable because they are negatively correlated with the inter-annotator agreement score. Results also show that in-context PVI estimates from larger models such as GPT3-175B are positively correlated with the inter-annotator agreement score, which indicates high reliability.
>
> To summarize, in-context PVI estimates made by larger models are reliable and they better capture information useful for making correct predictions. However, it is prohibitively costly to obtain their original PVI estimates. While it is feasible to obtain original PVI estimates by fine-tuning smaller models, our evidence has shown that in-context PVI estimates made by smaller models are less reliable. Thus, we do not include a section in our paper that directly compares in-context PVI estimates with the original PVI; however we do provide an indirect comparison in Section 5.4. We will make this reasoning clear in the revised version of the paper.
>
> In Section 5.3, we reported the results comparing the noisiness of in-context PVI with GPT2-125M and GPT3-175B. In the revised version of the paper, we will include more results with other models we experimented with (i.e., GPT-neo-1.3B, GPT-neo-2.7B, GPT-J-6B, GPT-JT-6B, Alpaca-7B, and Alpaca-13B) in the appendix for a more comprehensive comparison.
>
> **Reviewer1-Question-A: Are there other natural language descriptions for Prompt? Would it be helpful to include a more detailed task description?**
>
> Prompt description: Our paper includes prompt examples in Table 12 and 13 in Appendix A.2. We used two kinds of prompts in the calculation of in-context PVI: an *input-target* prompt and a *null-target* prompt. An *input-target* prompt has the following form:
> ```
> CONTEXT: I’m glad I saw anybody.
> QUESTION: Is this (0) unacceptable, or (1) acceptable?
> ANSWER: 0
> ```
> and a *null-target* prompt looks like the following:
> ```
> ANSWER: 0
> ```
> Detailed task description: We agree that it would be helpful to include a more detailed task description. We currently have a short paragraph in Section 4 that briefly describes the seven tasks with citations to the primary sources. Further descriptions and sample instances are provided in Tables 9-11 in Appendix A.1. We will add a more detailed description of each task in the appendix.
>
> **Reviewer1-Question-B: The current strategy is to select exemplar randomly, is there any plan to improve?**
>
> There are some demonstration selection methods in the current literature (such as in [1], [2], and [3]), and we agree that the quality of in-context PVI estimates may be improved if a more sophisticated method of selecting exemplars is applied. Our aim in the current study is to demonstrate that in-context PVI estimates show good qualities even with randomly selected exemplars. Furthermore, we set to develop in-context PVI and compare it with the original PVI. Experimenting with randomly selected exemplars is an unbiased approach. Exemplar selection is an active area of research with many variables to consider (e.g., avoiding bias, over-representing patterns). Thus, we leave the research on the combination of in-context PVI with other exemplar selection methodologies as the next step.
>
> [1] Li, X., & Qiu, X. (2023). Finding supporting examples for in-context learning. arXiv preprint arXiv:2302.13539.
> [2] Chang, T. Y., & Jia, R. (2022). Careful Data Curation Stabilizes In-context Learning. arXiv preprint arXiv:2212.10378.
> [3] ​​Zhang, Y., Feng, S., & Tan, C. (2022). Active example selection for in-context learning. arXiv preprint arXiv:2211.04486.

---

### Meta-Review · Area_Chair_bGRX · 2023-09-17

**Recommendation:** 4

**Metareview:**

The paper introduces in-context PVI as an efficient alternative to traditional PVI calculations, particularly beneficial for LLMs without the need for fine-tuning, and measures PVI only through prompting. The empirical analysis provides insights into the reliability and potential applications of in-context PVI.

Reviewers appreciate the efficiency and novelty of the in-context PVI method and the novel generalization of v-usable information from fine-tuning to in-context learning, which is seen as a valuable contribution. The comprehensive empirical analysis conducted in the study adds substantial credibility to the concept and establishes its reliability. The reviewers also express that the empirical results indicate a positive correlation between in-context PVI and accuracy, as well as inter-annotator agreement, suggesting its potential as a replacement for PVI.

However, the reviewers express concerns about the lack of a direct comparison between in-context PVI and traditional PVI. I agree with the suggestions by reviewers to directly measure the correlation between these two methods and report statistical significance, which would provide a more robust evaluation. I also suggest the authors to have a more in-depth discussion of the relationship between the two measures.

Overall, the paper offers an innovative and efficient approach to PVI calculation and presents valuable empirical insights. I suggest authors addressing the concerns regarding method comparison and adding statistical significance for experimental results to strengthen the paper's quality.

---

### Decision · Program_Chairs · 2023-10-07

**Decision:**

Accept-Findings

**Comment:**

The paper introduces in-context PVI as an efficient alternative to traditional PVI calculations, particularly beneficial for LLMs without the need for fine-tuning, and measures PVI only through prompting. The empirical analysis provides insights into the reliability and potential applications of in-context PVI.

Reviewers appreciate the efficiency and novelty of the in-context PVI method and the novel generalization of v-usable information from fine-tuning to in-context learning, which is seen as a valuable contribution. The comprehensive empirical analysis conducted in the study adds substantial credibility to the concept and establishes its reliability. The reviewers also express that the empirical results indicate a positive correlation between in-context PVI and accuracy, as well as inter-annotator agreement, suggesting its potential as a replacement for PVI.

However, the reviewers express concerns about the lack of a direct comparison between in-context PVI and traditional PVI. I agree with the suggestions by reviewers to directly measure the correlation between these two methods and report statistical significance, which would provide a more robust evaluation. I also suggest the authors to have a more in-depth discussion of the relationship between the two measures.

Overall, the paper offers an innovative and efficient approach to PVI calculation and presents valuable empirical insights. I suggest authors addressing the concerns regarding method comparison and adding statistical significance for experimental results to strengthen the paper's quality.